# Positive Effects of an Online Workplace Exercise Intervention during the COVID-19 Pandemic on Quality of Life Perception in Computer Workers: A Quasi-Experimental Study Design

**DOI:** 10.3390/ijerph19053142

**Published:** 2022-03-07

**Authors:** Sara Moreira, Maria Begoña Criado, Maria Salomé Ferreira, Jorge Machado, Carla Gonçalves, Filipe Manuel Clemente, Cristina Mesquita, Sofia Lopes, Paula Clara Santos

**Affiliations:** 1ICBAS, Instituto de Ciências Biomédicas Abel Salazar, Universidade do Porto, 4099-002 Porto, Portugal; jmachado@icbas.up.pt; 2ESS IPVC, Escola Superior de Saúde, Instituto Politécnico de Viana do Castelo, 4900-314 Viana do Castelo, Portugal; salomeferreira@ess.ipvc.pt; 3CBSin—Center of BioSciences in Integrative Health, 4000-105 Porto, Portugal; mbegona.criado@ipsn.cespu.pt; 4TOXRUN—Toxicology Research Unit, University Institute of Health Sciences, CESPU, 4585-116 Gandra, Portugal; 5UICISA: E—Health Sciences Research Unit: Nursing, Nursing School of Coimbra (ESEnfC), Portugal School of Health, Polytechnic Institute of Viana do Castelo, 4900-314 Viana do Castelo, Portugal; 6LABIOMEP—Laboratório de Biomecânica do Porto, Universidade do Porto, 4200-450 Porto, Portugal; 7ESDL IPVC, Escola Superior Desporto e Lazer, Instituto Politécnico de Viana do Castelo, Rua Escola Industrial e Comercial de Nun’Álvares, 4900-347 Viana do Castelo, Portugal; carlagoncalves@esdl.ipvc.pt (C.G.); filipe.clemente5@gmail.com (F.M.C.); 8Research Center in Sports Performance, Recreation, Innovation and Technology (SPRINT), 4960-320 Melgaço, Portugal; 9Delegação da Covilhã, Instituto de Telecomunicações, 1049-001 Lisboa, Portugal; 10ESS PPorto—Departamento de Fisioterapia, Escola Superior de Saúde, Politécnico do Porto, 4200-072 Porto, Portugal; ctmesquita@ess.ipp.pt (C.M.); srl@ess.ipp.pt (S.L.); paulaclara@ess.ipp.pt (P.C.S.); 11CIR—Centro de Investigação e Reabilitação, ESS P, 4200-072 Porto, Portugal; 12CESPU—Departamento de Tecnologias de Diagnóstico e Terapêutica, Escola Superior de Saúde do Vale do Sousa, Instituto Politécnico de Saúde do Norte (IPSN), 4585-116 Paredes, Portugal; 13CIAFEL—Centro de Investigação em Atividade Física, Saúde e Lazer, Faculdade de Desporto, Universidade do Porto, 4200-450 Porto, Portugal

**Keywords:** SF-36v2, telework, online exercise programme, occupational health

## Abstract

Computer workers’ sedentary work, together with less active lifestyles, aggravated by the COVID-19 pandemic, represents a high risk for many chronic diseases, leading to a decrease in health-related quality of life (QoL). Workplace exercises consist of a set of physical exercises, implemented during work breaks, that have multiple benefits for workers’ health. Aim: To assess the impact of online workplace exercises on computer workers’ perception of quality of life. Methods: Quasi-experimental study with two groups: a control group (n = 26) and an intervention group (n = 13). The inclusion criteria were that participants must be aged between 18 and 65 years old and the exclusion criteria included diagnosis of non-work-related medical conditions. The interventions consisted of workplace exercises, which were applied for 17 consecutive weeks, each session lasting 15 min, three times a week. The exercise programme, performed online and guided by a physiotherapist, consisted of mobility exercises, flexibility and strength exercises, with the help of a TheraBand^®^ for elastic resistance. The control group were not subjected to any intervention. A socio-demographic questionnaire and the Health Survey Questionnaire (SF-36v2) were used in two assessment stages (M0—baseline and M1—final of intervention). A mixed ANOVA with interaction time*group was used to evaluate the effect of the exercise programme. Results: A good perception of the QoL was obtained in both stages. The exercise programme had a positive effect in the domains of Pain (p_time*group_ = 0.012, η^2^_p_ = 0.158), Physical Function (p_time*group_ = 0.078, η^2^_p_ = 0.082), Physical Performance (p_time*group_ = 0.052, η^2^_p_ = 0.098), and Emotional Performance (p_time*group_ = 0.128, η^2^_p_ = 0.061). Conclusion: After 17 weeks of workplace exercises, it became clear that the intervention group positively increased their QoL perception, with this improvement being significant in the Pain domain, which resulted in an improvement in their health condition. Therefore, further studies are needed to determine the optimal exercise for CWs, with detailed exercise types, different intensities and focused on various health conditions.

## 1. Introduction

Currently, computers are a key element in the daily work of many individuals, representing the most common type of work in Europe, affecting millions of workers [1]. Although the number of individuals who use computers is constantly growing, this does not imply a direct relationship with health improvement; on the contrary, their use entails risks for workers’ health [2,3]. Office workers, or computer workers (CW), are a group who work for approximately two-thirds of their working hours on primary tasks that generally involve the use of computers, participating in meetings, presentations, reading and speaking on the phone in a sitting position [4,5,6]. These workers are, therefore, at increased risk of a number of chronic diseases due to their sedentary behaviour [7,8], and they are also at high risk of lowering their health-related quality of life (QoL) [9]. This sedentary work leads to poor postures, flexion postures adopted during long periods of work, associated with repetition of tasks, causing greater tension in muscle and ligament structure, thus, increasing the prevalence of work-related musculoskeletal injuries [5,7,10,11,12,13,14,15]. According to several studies, the cervical and lumbar regions are the most affected, followed by disorders in at least one region of the upper limbs [2,3,5,16,17,18,19,20,21]. Bragatto et al. (2016) reported a prevalence of musculoskeletal pain among computer workers of 55–69% [22], with spinal pain (cervical and lumbar regions) ranging from 31–54% and upper limb disorders between 5–15% in a Brazilian group [22]. These figures contribute to a mounting socioeconomic burden. For instance, in 2013, low back pain and neck-shoulder pain were reported to account for the third-highest amount of health-care spending in the United States [23].

In addition, an active lifestyle is described as improving overall health, decreasing the risk of chronic disease and increasing QoL [7]. According to Mayer et al. (2008), resistance exercises, or other forms of strength training, can improve a person’s capacity to support bone and cartilage, through improved musculature supporting movement around a joint, with potential to relieve stiffness and bringing about some pain relief [24].

The World Health Organization (WHO) defines health-related quality of life as the individual’s perception of their position in life, in the context of the cultural and value systems in which they are inserted, and in relation to their goals, expectations, standards and concerns [25]. There are also other factors that influence the QoL of these professionals, including considerable mental effort, short deadlines for delivering work, high levels of responsibility and intense exposure to computers [26]. The literature suggests that participation in ergonomics and on-site physical therapy treatment in the workplace (e.g., education, exercise and manual intervention) has a positive effect on decreasing work-related musculoskeletal injuries [27].

Physical therapy is considered an effective and economical tool that can be important to reduce the prevalence of work-related musculoskeletal injuries and increase worker productivity, through the implementation of specific physical exercise programmes [27,28,29]. Workplace exercise, or Labour Gymnastics, is defined as the performance of physical exercises, during sessions of 10 to 20 min, performed in the workspace and in line with the functions of each worker. These exercises are performed with the objective of increasing the strength and flexibility of the most demanded muscles during working hours, as well as enhancing the social integration and quality of life of workers [26,29,30]. As such, the role of the physiotherapist becomes essential, improving workers’ quality of life and preventing and/or treating musculoskeletal injuries [31].

Holzgreve et al. (2018) stated that workplace exercise programmes had numerous benefits, in terms of range of motion, and were effective in reducing musculoskeletal disorders and improving the quality of life in the workplace. In addition to being easily accessible and allowing simultaneous performance by several participants, they also help to minimize the risks associated with the use of new technologies, specifically from computers, improving postures with variability of movement, stretching the shortened muscles of the sitting posture and mobilising all the joints [19].

In December 2019, COVID-19 appeared, a communicable disease caused by infection of SARS-CoV-2 coronavirus [32]. This health condition spread, affecting several countries, leading the WHO to declare a pandemic that continues today. As such, in order to prevent transmission, mandatory teleworking was ordered, which meant that all face-to-face activities were carried out online [33]. This meant that workers’ homes had to become a place of work, education and leisure. Teleworking has several benefits, such as a better balance between family and work, reduced fatigue and increased productivity [34]. However, on the other hand, the blurring of physical and organisational boundaries between work and home, extended working hours and limited support from organisations, can negatively impact the physical and mental health of workers [34], leading to a decrease in health-related quality of life [35,36]. In this scenario, Labour Gymnastics will be carried out remotely, through digital videoconference tools. Recent reviews investigating the effects of physical exercise on the health of office workers [21,37,38] reported the significant and protective effects of physical exercise on musculoskeletal pain symptoms (i.e., neck pain and low back pain) [21,29,38], with some studies indicating a significant association between physical exercise and QoL [21,29,31,39]. Previous reviews focused mostly on office workers with musculoskeletal disorders, the most common work-related chronic conditions. However, little is known about the relationship between physical exercise and QoL in healthy office workers, engaging in sedentary behaviour during most of their working day, for whom the risk of many chronic diseases remains high, or QoL in office workers with other types of work-related diseases [26,31,38]. Recent studies carried out during the pandemic also suggested that regular physical activity, albeit via online programming, can be an accessible auxiliary tool for the immune system against possible COVID-19 infection [40].

Improving QoL is, therefore, a challenge for occupational health, not least because the work context is crucial for the development and creation of health promotion actions within the scope of work activity [38,39,41]. As far as we know, there are few studies about the effects of exercise intervention on the perception of health-related quality of life in computer workers, but studies of this kind would facilitate more efficient preventive and interventional approaches. In this context, it is urgent to investigate the perception of QoL that those workers have and how to change attitudes, in order to introduce successful programmes for the promotion of health-oriented physical activity, in a variety of social groups [39,42,43]. Thus, the aim of the present study was to assess the impact of online workplace exercises on the perception of QoL in CWs. As such, the study hypothesis was that online workplace exercise leads to an increased perception of QoL in CWs.

## 2. Materials and Methods 

### 2.1. Study Design and Ethics

A quasi-experimental study was performed with an intervention group and a control group. The sample selection was considered for convenience as it was a non-probability sampling and the sample consisted of participants who decided to become part of the research. This approach was adopted because we had to select the sample members solely from available and easily accessible participants. An online workplace exercise programme was carried out by a physiotherapist, through Microsoft Teams software (Microsoft Corporation^®^, Albuquerque, NM, USA). The exercise programme was applied to the intervention group for 17 consecutive weeks (from 7 April to 31 July 2021), each session lasted 15 min, three times a week. Data were collected from adult CW. Ethical approval for this study was obtained from the Ethics Committee at the Abel Salazar Institute of Biomedical Sciences CHUP/ICBAS (963). All the participants were informed about the study aims and procedures and they provided consent for their participation. The participants could refuse to participate in the study at any point under Law 67/98 of 26 October 1998 (Law on the Protection of Personal Data (transposing into the Portuguese legal system Directive 95/46/EC of the European Parliament and of the Council of 24 October 1995) on the protection of individuals with regard to the processing of personal data and on the free movement of such data) and the World Medical Association Declaration of Helsinki Ethical Principles for Medical Research.

### 2.2. Sample Recruitment and Elegibility Criteria

The target population of this study consisted of 462 adult CW from an automotive sector company in northern Portugal. 

Defined inclusion criteria were: aged between 18 and 65 years old and having signed the informed consent and all participants were full-time employees. The exclusion criteria included diagnosis of non-work-related medical conditions, such as: ankylosing spondylitis, chronic joint diseases, neurological diseases, relevant (osteoarticular) surgeries, significant artificial joint replacement, articulation, multiple sclerosis, myotonic dystrophy or neurodegenerative diseases and congenital malformations of the musculoskeletal system [19,31].

Prior to the study, a visit was made to the company to establish an initial contact with the possible participants and their work environment. The present paper was part of an occupational health study, so a more detailed diagnostic evaluation of the participants was provided in previous papers by our research team [44,45]. The effect of the exercise programme on quality of life was assessed through the responses of the participants to the questionnaire SF-36v2, validated for the Portuguese population [46]. Before completing the questionnaire, all the computer workers were informed by the company’s human resources department about the objectives of the study and the questionnaire was presented. Subsequently, time was allotted to clarify any doubts and questions that remained. Following clarification, the questionnaire link was provided to the potential participants via email by the company’s human resources department. The forms were filled out in two stages: a first one from 10 December 2020 to 29 January 2021 and a second from 15 to 31 July 2021. Of the 462 CWs in the target population, 337 were not considered eligible, 329 individuals did not respond to the questionnaires, 7 refused to participate and 1 was excluded for presenting a diagnosis of ankylosing spondylitis, such that a final sample of 125 CW was formed. The final sample was distributed non-randomly into two groups, the control group (CG) and the intervention group (IG). As indicated above, the sample selection was considered for convenience, as the intervention group consisted of 26 volunteer workers who agreed to participate in the online exercise programme and the remaining 99 workers were moved to the control group. Of the 26 participants in the intervention group at M0, 13 did not attend the complete programme and of the 99 volunteers in the control group at M0, 73 did not respond to the questionnaires. As such, at M1, the sample consisted of 13 participants in the intervention group and 26 participants in the control group (Figure 1).

### 2.3. Questionnaire for Collecting Data

The final questionnaire consisted of two parts: one with sociodemographic questions and a second with the Health Status Questionnaire—MOS Short Form Health Survey 36-Item v2 (SF-36v2). A pilot study was carried out to test the procedures. The survey was designed to take no longer than 15 min to complete and was a self-administered questionnaire.

#### 2.3.1. Sociodemographic Questionnaire

This questionnaire (Appendix A) was designed by the main researcher to characterise the sample and collect sociodemographic data [44]. Firstly, general information was included, such as sex, birth date, relationship status and education. Anthropometric variables, such as weight and height, were self-reported, followed by questions about medical history and lifestyle. Lastly, issues related to work were included (number of working hours per day, rotating or fixed shift, years of service, other paid employment in addition to the current job, and if so, how many hours per week in this extra activity, current employment status, type of employment contract and managerial functions). To capture additional contextual information, number of hours per week of working in domestic and leisure activities were included.

#### 2.3.2. MOS Short Form Health Survey 36-Item v2 (SF-36v2)

This questionnaire assesses general health status [46,47]. It contains 36 items, distributed over 8 dimensions: Physical Function, Social Function, Physical Performance, Pain, Mental Health, Emotional Performance, Vitality and General Health. Each dimension is scored from 0 to 100, with 100 representing the best health status and 0 indicating the worst health status [46]. The SF-36v2 questionnaire is translated and validated for the Portuguese population and has good internal consistency, with a Cronbach’s Alpha ranging between 0.60 for Social Function and 0.87 for Physical Function and General Health [46,47]. In terms of reproducibility (1 week), it has an r ranging between 0.45 for Pain and 0.79 for Physical Performance and a alpha coefficient ranging from 0.45 for Mental Health and General Health to 0.84 for Pain [46,47]. Regarding content validity, it passes internal consistency tests with overall success between 90% and 100% and discrimination tests with overall success rates from 56% to 100% [46,47].

### 2.4. Procedures

#### 2.4.1. Pilot Study

An exercise programme was designed by the main researcher and approved by experts (external to the organisation): three specialist physiotherapists, a rehabilitation nurse and an exercise physiologist. This programme was based on the literature [48,49,50,51] and tested through a pilot study, performed online through the Microsoft Teams software platform (Microsoft Corporation^®^, Albuquerque, NM, USA) to analyse the need for possible changes before being applied to the intervention group. Adjustments were made in the order of exercises to apply the final exercise programme version (Appendix A).

#### 2.4.2. Data Collection

Data collection was performed in two separate stages. The first stage (M0) began on 10 December 2020 when the company’s human resources department sent the link to fill in the questionnaires by email. The questionnaires were prepared and answered using Google Forms online software (Google, Mountain View, CA, USA) and were available for completion until 29 January 2021. These questionnaires allowed us not only to assess the selection criteria, but also to collect information about sociodemographic aspects and the health condition of the participants. The filling in of the questionnaires also made it possible to distribute the participants into each group (control group and intervention group), according to their decision. The second stage (M1) occurred after 17 weeks of the intervention, from 15 July to 31 July 2021, with a new application of the questionnaires, using the previous methods and under the same conditions.

#### 2.4.3. Intervention

The intervention consisted of the implementation of an online exercise programme through Microsoft Teams software (Microsoft Corporation^®^, Redmond, WA, USA). The exercises were carried out in the IG during working hours and supervised by the physiotherapist specialising in occupational health, with over 10 years of experience. The application of the exercise programme began on 7 April 2021 and was performed three times per week on alternating days. The timetable was defined by the company (at 9.15 am and at 1.45 pm) for 17 consecutive weeks. Each session lasted 15 min. Based on the principles of muscle training with progressive resistance [52], each exercise cycle started with a pretensioned rubber band, increased progressively following the principle of periodisation and progressive overload. The loads were applied by rubber bands measuring 20 cm wide by 2 m long with colour-coded graduations (TheraBand^®^, Akron, OH, USA) [53]. The exercise programme was standardised for all the participants, with no differences between men and women. The intervention protocol consisted of 7 mobility exercises, 8 flexibility exercises and 7 strength exercises with the help of a TheraBand^®^ brand elastic resistance band. This strength was green in colour, which reflects approximately 2.3 kg at 100% elongation. This was the colour chosen for resistance, as it is considered to be of an intermediate level [54]. Each exercise was performed in 1 set of 8 repetitions with 10 seconds’ pause between each cycle of exercises (Appendix A). After 3 weeks, tolerance to the next level of resistance was assessed and all the participants were asked to try another three levels of resistance using the blue (2.6 kg); black (3.3 kg) and silver (4.6 kg) TheraBands (Akron, OH, USA).

The physiotherapist focused on positive comments to maintain motivation and programme compliance. All participants allocated in the CG continued performing their normal daily activities. During the evaluation controls performed at the start of any structured exercise or physical activity programme, as well as individual or team sport activities, they could follow the programme but would be excluded from the research.

To increase programme membership and achieve greater health gains in this community, the company asked the research team to lead a webinar held on 7 April to celebrate World Health Day. The main objective of this education and promotion session was to actively involve each employee in their health project. To maintain the motivation of the participants, the physiotherapist was available for any queries at all times. After a month and a half of the intervention, a satisfaction questionnaire and improvement proposals were presented, to be completed by the IG participants, for adjustments or improvements to future Labour Gymnastics exercise programmes.

#### 2.4.4. Data Processing

The answers to the questionnaires were exported to Microsoft Excel software, version 16.0 (Microsoft Corporation^®^, Redmond, WA, USA). To facilitate the reading of the data from the SF-36v2 questionnaire, the Raw Scale calculation was used, in which the scores obtained were converted to a scale from 0 to 100 (worst and best health status, respectively).

Subsequently, these data were transferred to the IBM SPSS Statistics^®^ program, version 27.0 (Statistical Package for the Social Sciences^®^, IBM Corp., Armonk, NY, USA) for statistical analysis of the data [55].

### 2.5. Statistics

Descriptive and inferential statistics were made considering a significance level of 5%. Descriptive statistics were used to characterise the sample through measures of central tendency, including the mean and dispersion measures, such as the standard deviation. Absolute frequencies and relative frequencies were used to characterise the sample in terms of age, gender, obesity, management position, number of breaks and hours worked per day.

The Student’s *t*-test and the Fisher’s exact test were used for the comparison of the two groups before the exercise programme (M0). To assess the effect of the exercise programme in the QoL domains, a mixed ANOVA with interaction time*group was used. Partial Eta squared (η^2^_p_) was used to assess the effect size of the interaction (η^2^_p_ = 0.01 small effect, η^2^_p_ = 0.06 medium effect, η^2^_p_ = 0.14 large effect) [56]. The mixed ANOVA was followed by a Student’s *t*-test for the comparison M0–M1 within each group. The effect size of the differences before–after the programme within each group was evaluated with Cohen’s d [56] (d = 0.20 small effect, d = 0.50 medium effect, d = 0.80 large effect). 

## 3. Results

### 3.1. Sample Characteristion

Table 1 shows the sample characteristics (IG, CG and total) before applying the exercise programme (M0).

As can be seen in Table 1, the questionnaire was answered by 39 CW (13 of the IG and 26 of the CG), aged between 25 and 50 years old (M = 36.1, SD = 6.6), mostly men (64.1%). The two groups did not differ significantly with regard to age (*p* = 0.695) or gender (*p* = 0.157). Despite the differences in gender not being statistically significant, the percentage of women who participated in the programme (53.8%) was higher than the percentage of men (46.2%). This may be explained because females had higher compliance with the online Exercise Programme than males.

Out of the 39 workers, 17.9% were obese (15.4% in the IG, 19.2% in the CG, *p* = 1.000). 

As for the variables related to work, the majority of the participants worked 8 or more hours per day (100.0% in the IG, 88.5% in the CG, *p* = 0.538), did not have a management position (69.2% in the IG, 61.5% in the CG, *p* = 0.733) and reported fewer than three daily working breaks (84.6% in the IG, 61.5% in the CG, *p* = 0.269).

### 3.2. Effect of the Exercise Programme on Quality of Life (SF-36)

Table 2 show the results of the comparison of the eight domains of SF-36, before (M0) and after (M1) the exercise programme within each group. 

The analysis of the interaction time*group shows a large effect of the Exercise Programme in the domain of Pain (*p* = 0.012, η^2^_p_ = 0.158), with an increase in the IG’s mean, from 79.0 (SD = 17.1) to 87.4 (SD = 12.7) (*p* = 0.121, d = 0.56), and a decrease in the CG, from 79.2 (SD = 19.0) to 71.1 (SD = 20.1) (*p* = 0.035 d = 0.42) (Table 2 and Figure 2).

Despite not being statistically significant, the effect size of the interaction term was medium in the domains Physical Function (*p* = 0.078, η^2^_p_ = 0.082), Physical Performance (*p* = 0.052, η^2^_p_ = 0.098) and Emotional Performance (*p* = 0.128, η^2^_p_ = 0.061). Regarding Physical Function, the differences from M0 to M1 within each group show a small increase in the IG’s mean (from 93.8 to 95.0, *p* = 0.610, d = 0.16) and a decrease in the CG (from 94.4 to 88.1, *p* = 0.027, d = 0.54). As for Physical Performance, the mean increased in the IG from 70.2 (SD = 23.5) to 80.3 (SD = 13.0) (*p* = 0.204, d = 0.53) and decreased in the CG from 81.3 (SD = 16.6) to 77.5 (SD = 16.2) (*p* = 0.246, d = 0.23). A similar trend was found concerning Emotional Performance: an increase in the IG (*p* = 0.203, d = 0.42) and a decrease in the CG (*p* = 0.588, d = 0.10) (Table 2 and Figure 2).

The effect of the Exercise Programme was small or negligible in the domains General Health (p_time*group_ = 0.598, η^2^_p_ = 0.008), Vitality (p_time*group_ = 0.404, η^2^_p_ = 0.019), Social Function (p_time*group_ = 0.663, η^2^_p_ = 0.005) and Mental Health (p_time*group_ = 0.350, η^2^_p_ = 0.024) (Table 2). 

Figure 2 shows the scores (means by group and score of each individual) of the four SF-36 domains (Pain, Physical Function, Physical Performance and Emotional Performance) in which the exercise programme had a medium to large effect (effect size of the interaction higher than 0.060).

## 4. Discussion

The aim of this study was to understand the effects of an online supervised workplace exercise intervention programme on computer workers’ perception of quality of life. 

It was found that at both assessment stages, computer workers had a good perception of quality of life, obtaining means above 50 in all dimensions, values of QoL considered in the literature as acceptable [57]. As evidenced by a number of studies, good work environment conditions directly influence QoL in computer workers. In this regard, in line with the measures implemented during the pandemic, they all worked in clusters, with weekly alternations between teleworking and working at the company. The company has a good working environment: good lighting and low noise, adapted to the needs of each worker, two factors considered protective for the prevention of chronic diseases. Another factor that may influence these results is the fact that all the participants were young adults (mean age: 36.1). These conditions may explain the relatively high values of QoL obtained in our sample at M0.

In general, the exercise programme showed positive effects, as the perception of quality of life tended to increase in the IG, which is in agreement with the literature [24,25,27,43,52,58].

The analysis of the differences from M0 to M1 within each group showed a large effect of the Exercise Programme in the Pain domain, which increased its values significantly, when comparing the intervention group at M0 with M1. The results in the Pain domain increased, which means that pain levels decreased. Ware et al. [59] indicated that high values obtained in the Pain domain indicate that the person has no pain or limitation and low values obtained in the Pain domain indicate very intense and extremely limiting pain. In other words, higher scores in the SF-36v2 questionnaire for the Pain dimension indicate lower pain symptoms, representing better health status [58]. As such, the intervention group had a higher score in the Pain domain, resulting in improvement after the exercise programme. These results were expected, in line with the literature, as they were also seen in studies carried out with specific exercises for pain, where the intervention group had higher Pain domain values, resulting in improvements after an exercise programme [29,60,61,62]. Sousa et al. [63], in an exercise programme lasting two months and performed three times a week, reported improvements in muscle pain and fatigue. Furthermore, exercises performed during working hours, even if performed for a short period of time, can contribute to a reduction in stress and to relaxation on the part of the participants [21,64,65]. In addition, according to Booth et al. [65], physical exercise has an impact on pain reduction, which consequently improves the participant’s physical performance and psychological status. This is due to exercise-induced analgesia, as well as structural and functional adaptations that occur in the brain [65]. Gobbo et al. [21] found a marked reduction in pain after a supervised intervention exercise programme in office workers. In fact, exercising under the supervision of a professional instructor ensured correct postures and resulted in strong exercise intensity. The same study indicates that the best improvement was recorded in supervised exercise programmes and in video-supported protocols performed in the workplace. The effect may be generated with short duration sessions during the working day, with only 10–15 min of adapted exercise to be performed 3–5 days per week [21]. It is important to point out that diverse exercise types with varying intensities can be adopted in the exercise programme. Therefore, further studies are needed to determine the optimal exercise for office workers with different health conditions.

When evaluating other dimensions of quality of life, before and after exercise intervention, it was observed that Physical Function, Physical Performance and Emotional Performance were the dimensions most affected. In this study, despite not being statistically significant, a medium increase was observed in the effect size of the interaction term in the domains Physical Function, Physical Performance and Emotional Performance. As such, there seems to be a tendency for the exercise programme to have a positive effect on the perception of QoL. In other words, the exercise programme was positive for these workers, as the perception of quality of life tended to increase. These results are in agreement with various studies, which indicated a strong association between exercise interventions and improvements in QoL in office workers, and significant and positive effects of physical exercise on QoL [7,19,29,39,48,66,67,68,69,70,71,72,73,74]. Various studies also reported that a significant improvement in QoL was observed in office workers who performed supervised physical exercise [21,29,30,39,48,50,57,61,68,75].

The same trend was observed in the Emotional Performance score, which increased significantly in the IG. This finding is also in accordance with other studies [29,48,68,76].

The effect of the Exercise Programme was small or negligible in the domains General Health, Vitality, Social Function and Mental Health. Regarding Vitality and Social Function, although these dimensions showed improvements after the intervention, this increase was also reflected in the control group. As such, it is not possible to infer that the improvements in the intervention group are due to Labour Gymnastics. However, according to the literature, Labour Gymnastics provide an increase in physical and mental health, resulting in improved quality of life. Thus, physically active people are more content and more alert [31]. There is evidence in the literature that exercise is beneficial for mental health; it reduces depression, negative moods and anxiety, and improves self-esteem and cognitive functioning [77].

These results are in agreement with Nguyen et al. [39], who suggested that supervised physical exercise significantly improved general QoL, although mental health was improved only by unsupervised physical exercise interventions [29,39]. These authors also suggest that further studies are needed with supervised and unsupervised interventions.

Taking into account what was described in the discussion, it appears that the presence of a physical therapist in the company was beneficial, since the values of perceived quality of life at M0 were already high. This can be explained by the fact that the physiotherapist has been at this company since 2016. This professional is a key player in this environment, given that, due to his knowledge in the areas of ergonomics and biomechanics, he can intervene as part of a multidisciplinary team to improve well-being and prevent and treat musculoskeletal injuries among workers [31]. In addition, the supervised exercise programme increased the perception of quality of life in the CWs, as shown by the values obtained at M1 in the intervention group.

### Study Limitations

In the performance of this study, some limitations were also found, such as the small sample size and the fact that the gender was mostly male, which led to heterogeneity between the groups. Furthermore, on one hand, the sample was formed of volunteers, which produced a selection bias. In addition, out of a total of 462, only 125 CWs answered the questionnaires, representing a non-response bias. It was also not possible to verify whether the workers answered the questionnaire themselves as it was answered online. Another important limitation of this study was the discrepancy between groups: 13 individuals in IG versus 26 in CG, with a 10.4% rate of adherence to the exercise programme, which could be explained by varying motivation levels, attributable to organisational issues, personal reasons or feelings of embarrassment about exercising in public or with colleagues [21,78]. As such, many personal variables, including physiological, behavioural and psychological factors, may influence motivation to join physical activity programmes. Thus, to make physical activity part of the daily life of CWs, it will be important to understand common barriers to physical activity and to create strategies to overcome them. The high perception of QoL at the beginning of the study could also influence the motivation to join the programme, resulting in unwillingness to participate and, thus, a low participation rate. To address these limitations in future studies, it is recommended that a larger sample be used, consisting of computer workers from several companies. Lastly, the intervention time of only 17 weeks represented another limitation. This time was considered to be too limited to obtain significant improvements in the results. However, Lima et al. [79] concluded that a three-month workplace programme of physical exercises might contribute to improving flexibility and mobility; by participating in proposed sessions two, three and five times a week, all the groups demonstrated statistically significant improvements in mobility and flexibility [79].

It would also be important to monitor these workers in exercise programmes for a longer time, in order to understand whether more significant results would appear and to make this physical exercise a habit during their breaks [62]. In this regard, the opinion of the participants, expressed throughout satisfaction questionnaires, could be of great importance to adapt the exercise programme to be more attractive for future participants.

## 5. Conclusions

The main strength of this study was evaluating the impact of an online workplace exercise intervention on the perception of quality of life of computer workers.

We conclude that the intervention group positively increased their QoL perception, as improvements in Pain, Physical Function, Physical Performance and Emotional Performance domains were found. The improvement was significant only in the Pain domain. These findings are important, as they can be used to adapt the promotion of Physical activity in the workplace, reflecting a positive perception of health, related to these workers’ QoL.

Further studies are needed to determine the optimal exercise programme for CWs, taking into account aspects, such as exercise types, different intensities, with or without supervision and a focus on various health conditions [39].

In this context, further analysis is needed to confirm the observed results, but there seems to be a clear need to direct efforts and resources towards improving computer workers’ QoL by boosting CWs’ motivation to join exercise programmes in the workplace and implementing suitable exercises programmes and policies, with the consequent social, economic and environmental impacts on physically active, healthy populations.

## Figures and Tables

**Figure 1 ijerph-19-03142-f001:**
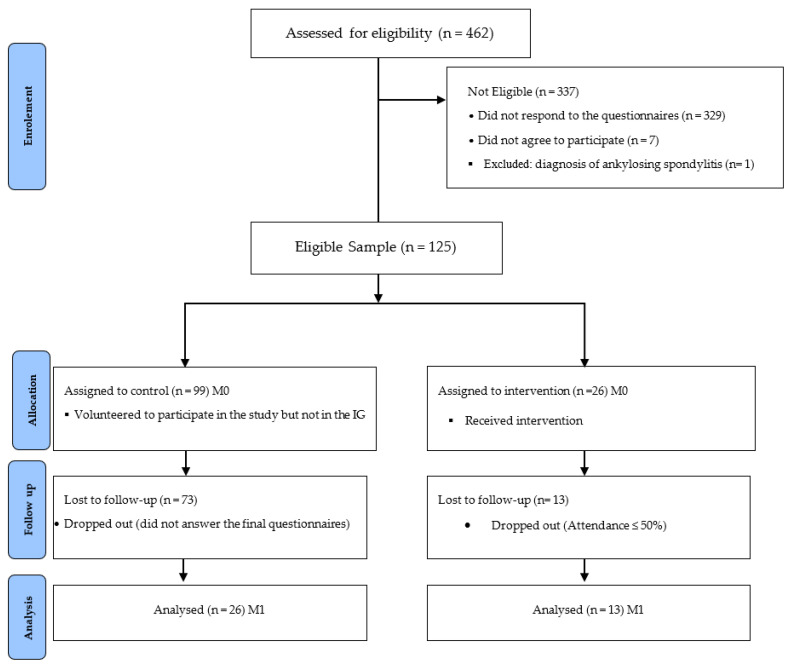
Flow diagram of the recruitment process. M0—baseline, M1—final of intervention, IG—Intervention Group.

**Figure 2 ijerph-19-03142-f002:**
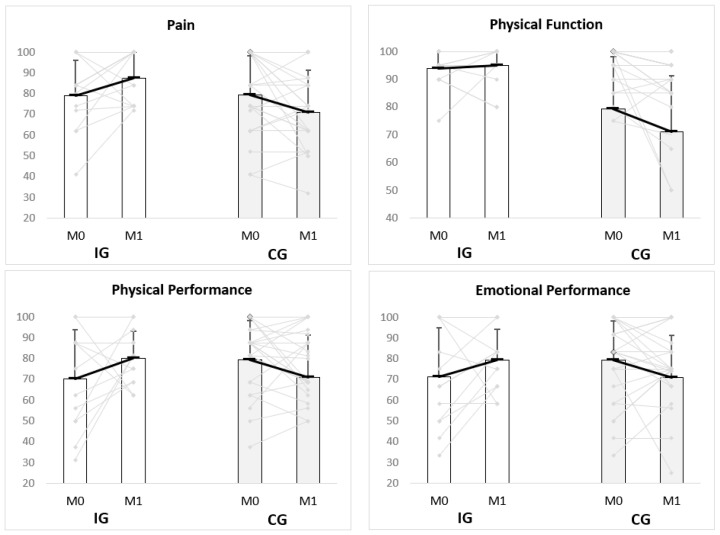
Means (SD) and scores of each individual for the domains Pain, Physical Function, Physical Performance and Emotional Performance of SF-36 (score from 0 to 100, with 100 representing the best health status and 0 indicating the worst health status) in M0 and M1, by group (the black lines represent the group mean and the grey lines the values for each individual). M0—baseline, M1—final of intervention, IG—Intervention Group, CG—Control Group.

**Table 1 ijerph-19-03142-t001:** Sample characteristics in M0 regarding sex, age, obesity, daily working hours, management position, and daily working breaks.

	IG (n = 13)	CG (n = 26)	Total (N = 39)	*p*-Value
Age				
Mean (SD)	35.5 (5.1)	36.4 (7.4)	36.1 (6.6)	0.695 ^(1)^
Gender				
Female—n (%)	7 (53.8%)	7 (26.9%)	14 (35.9%)	0.157 ^(2)^
Male—n (%)	6 (46.2%)	19 (73.1%)	25 (64.1%)	
Obesity				
No—n (%)	11 (84.6%)	21 (80.8%)	32 (82.1%)	1.000 ^(2)^
Yes—n (%)	2 (15.4%)	5 (19.2%)	7 (17.9%)	
Daily working hours				
Mean (SD)	8.5 (0.8)	8.3 (0.7)	8.4 (0.7)	0.199 ^(1)^
<8 h—n (%)	0 (0.0%)	3 (11.5%)	3 (7.7%)	0.538 ^(2)^
≥8 h—n (%)	13 (100.0%)	23 (88.5%)	36 (92.3%)	
Management position				
No—n (%)	9 (69.2%)	16 (61.5%)	25 (64.1%)	0.733 ^(2)^
Yes—n (%)	4 (30.8%)	10 (38.5%)	14 (35.9%)	
Daily working breaks				
Mean (SD)	1.8 (0.9)	2.0 (1.0)	2.0 (0.9)	0.550 ^(1)^
<3 breaks—n (%)	11 (84.6%)	16 (61.5%)	27 (69.2%)	0.269 ^(2)^
≥3 breaks—n (%)	2 (15.4%)	10 (38.5%)	12 (30.8%)	

IG—Intervention Group; CG—Control Group; SD—Standard deviation. ^(1)^
*p*-Value of Student’s t test; ^(2)^
*p*-Value of Fisher’s exact test.

**Table 2 ijerph-19-03142-t002:** Effect of the programme on the quality of life (SF-36) (IG: n = 13, CG: n = 26).

	SF-36 Domains ^(1)^	Group	M0 M (SD)	M1 M (SD)	*p*-Value ^(2)^ M0-M1	Cohen’s d	Interaction ^(3)^	
							*p*-Value	η^2^_p_
Physical Dimension	Physical Function	IG	93.8 (6.8)	95.0 (7.4)	0.610	0.16	0.078	0.082
		CG	94.4 (8.0)	88.1 (14.4)	0.027	0.54		
	Differences intergroups ^(4)^	*p*-value Cohen’s d	0.826 0.08	0.114 0.60				
	Physical Performance	IG	70.2 (23.5)	80.3 (13.0)	0.204	0.53	0.052	0.098
		CG	81.3 (16.6)	77.5 (16.2)	0.246	0.23		
	Intergroups differences ^(4)^	*p*-Value Cohen’s d	0.147 0.54	0.598 0.19				
	Pain	IG	79.0 (17.1)	87.4 (12.7)	0.121	0.56	0.012	0.158
		CG	79.2 (19.0)	71.1 (20.1)	0.035	0.42		
	Intergroups differences ^(4)^	*p*-Value Cohen’s d	0.971 0.01	0.012 0.97				
	General Health	IG	75.8 (13.3)	75.9 (13.3)	0.961	0.01	0.598	0.008
		CG	73.0 (15.0)	71.0 (16.2)	0.417	0.13		
	Intergroups differences ^(4)^	*p*-Value Cohen’s d	0.576 0.20	0.345 0.34				
Mental dimension	Vitality	IG	48.1 (20.6)	67.3 (15.6)	0.004	1.05	0.404	0.019
		CG	55.5 (17.1)	70.3 (18.4)	<0.001	0.83		
	Intergroups differences ^(4)^	*p*-Value Cohen’s d	0.238 0.39	0.615 0.18				
	Social Function	IG	79.8 (19.5)	80.8 (19.5)	0.794	0.05	0.663	0.005
		CG	76.4 (20.4)	80.2 (21.0)	0.366	0.18		
	Intergroups differences ^(4)^	*p*-Value Cohen’s d	0.625 0.17	0.930 0.03				
	Emotional Performance	IG	71.2 (24.0)	79.5 (14.7)	0.203	0.42	0.128	0.061
		CG	79.8 (20.3)	77.9 (18.9)	0.588	0.10		
	Intergroups differences ^(4)^	*p*-Value Cohen’s d	0.245 0.39	0.791 0.09				
	Mental Health	IG	63.8 (12.4)	80.4 (13.8)	0.001	1.26	0.350	0.024
		CG	68.1 (17.9)	80.0 (21.0)	<0.001	0.61		
	Intergroups differences ^(4)^	*p*-Value Cohen’s d	0.450 0.27	0.953 0.02				

IG—Intervention Group; CG—Control Group; M—Mean; SD—Standard deviation; η^2^_p_—partial Eta squared; ^(1)^ Score from 0 to 100, with 100 representing the best health status and 0 indicating the worst health status; ^(2)^
*p*-value of Student’s paired *t*-Test—comparison M0-M1 within each group; ^(3)^ interaction group*time (Mixed ANOVA); ^(4)^ comparison between groups in M0 and M1—Student’s independent *t*-test.

## Data Availability

The data presented in this study are available on request from the corresponding author. The data are not publicly available due to the accordance to the confidentiality agreement with the company does not allow the public disclosure of the data available.

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
