# Peer review of "Positive Effects of an Online Workplace Exercise Intervention during the COVID-19 Pandemic on Quality of Life Perception in Computer Workers: A Quasi-Experimental Study Design"

_ijerph, 2022, doi:10.3390/ijerph19053142_

Round 1

Reviewer 1 Report

This is an interesting study investigating the effects of an online detailed exercise program on various parameters of computer workers. Whereas the authors measured several variables, pain was significantly lower after the intervention and should be the focus of the paper. The authors conclude that the overall QOL went up after the intervention. Such conclusions should be moderates. Independent of the effect sizes, if any variable was not significant, the authors should avoid stating that it improved or not. Please tone down the language throughout the discussion section and focus only on the significant findings. 

Reviewer 2 Report

This manuscript reflects the important topic/issue of computer workers during the COVID-19 pandemic, and the impact of workplace exercise on their quality of life. The paper has the potential to add to the field of literature and offers a level of novelty, but there substantial modification is required before this can happen:

The Abstract is unclear and would benefit from being revised. Specifically, "brings" should be "bring", there is no need for "an" online workplace exercises. What does "no randomized were made" mean? What does "was not submitted" mean? What does "A socio-demographic questions" mean. The Abstract contains a lot of information about the study Methods, but there is no reference to the SF36.

Introduction

  • Paragraph 1, line 73: I am unsure what is meant by "less adequate postures"?
  • The second paragraph is brief, this would benefit from elaboration. Where in the world does the data from reference (19) come from. Is this the same globally? What are the impacts of such pain? How does an active lifestyle do this? Does an active lifestyle assisting with pain management and musculoskeletal issues?
  • Paragraph 3, line 86: what is meant by intense exposure to digital equipment?
  • In the initial paragraphs of the Introduction it becomes clear that a musculoskeletal is apparent. I think the authors could do a more precise job of capturing the relationship between musculoskeletal conditions, pain, and quality of life here.
  • Paragraph 5: how can GL programs minimize the risks associated with the use of new technologies?
  • Paragraph 6, line 106: what is meant by "quickly dragged on"? What is meant by "until today" - does that mean it ends today? (line 107).
  • Paragraph 6, line111: "better family and professional integration" - as opposed to what?
  • Line 115: I do not think there is a need for "inevitably".
  • The final five sentences in the Introduction can be merged together to make one paragraph, and possibly simplified to enable the background and aim to be clearer. What is meant by "it is indispensable to investigate"?

Methods

  • I recommend stating that a convenience sample was used for this study. Why was this approach adopted?
  • A major limitation of the method is that a "questionnaire" is referred to regularly, and with this seems to be an expectation that the reader knows about this questionnaire. What is the questionnaire, and what evidence exists that it is valid and reliable? How were the participants informed about the questionnaire?
  • Lines 157-162: were participants full-time?
  • Lines 163-166: it would be beneficial to include a table of further description of the participant diagnostics in this study.
  • The amount of participant dropout is significant. Rationale for this occurring would strengthen the Method. What was put in place to prevent this?
  • 2.3 Questionnaire: why is this not provided in supplementary materials?
  • 2.3.1, line 195: "heigh" is spelt incorrectly; should be height.
  • 2.4.1: were the experts external to the organisation, or part of the organisation?
  • Lines 260-261: if participants were invited to continue in performing their normal daily activities, what other options did they have?

Results

  • 3.2, line 320: "Despite not being statistically significant differences inter-groups" - I am unsure what this means.

Discussion

  • Paragraph 2, line358: participants are mentioned as being young. Are they? Young as opposed to who?
  • There is inconsistency throughout the Discussion with the spelling of "programmes" and "programs".
  • The explanation of pain values increasing is a positive result? Please be clear with how this is explained and articulated; pain levels improved but scores increased?
  • Line 392: I do not believe "it was also possible" is needed here.
  • Lines 397-403: consider making this all one paragraph.
  • Line 413: "who suggested" rather than "that suggested".
  • Lines 417-423: the explanation of the physical therapist being present in the company since 2016 needs to come earlier (Methods).
  • Lines 433-437: in explaining study limitations there is mention of adherence, motivation levels, organizational issues, personal reasons, or feeling embarrassed. These are important factors and warrant further explanation. Why were these factors influential?

Conclusions

  • Line 455: I encourage you to consider not using the terminology "It was possible".
  • Line 461: should state "into account aspects such as exercise types". 

Reviewer 3 Report

This is an interesting study on the effect of physical exercise (provided online) at workplace in the COVID-19 era on quality of life of computer workers and could be accepted for publications after the following revisions.

-line 73: Work-related musculoskeletal injuries : may be work should not be in capital letter

-too many abbreviations. Avoid using so many acronyms if possible in the Introduction and Discussions, at least.
-more details on COVID-19 and the benefits that physical activity, including that practiced outdoors,  could give in preventing the viral diseases and other disorders , should be provided in the introduction citing at least the work with DOI: 10.1007/s10311-021-01321-9

-give more details on 'Law 67/98 of 26 October 1998 ': country, place?

-line 178 onwards explain or repeat the explaination for M0, M1...

-check 'were possible to ', line 258. In general improve english level of section 2

-Table 1: I see differences in female/male ratios and other characteristics of participants in IG and CG. Comment more extensively on this already in section3

-provide a reference after 'occur in the brain', line 379

-line 395: I see a cancelled word (had), check

-line 426: please consider to provide Limitations as a separate section
